# Impact of Music Interventions on Depression in Care Home Residents with Dementia: UK Results from Music Interventions for Depression and Dementia in Elderly Care RCT

**DOI:** 10.3390/geriatrics10060166

**Published:** 2025-12-15

**Authors:** Justine Schneider, Joanne Ablewhite, Jodie Bloska, Martin Orrell, Helen Odell-Miller, Jorg Assmus, Christian Gold, Vigdis Sveinsdottir

**Affiliations:** 1Institute of Mental Health, University of Nottingham, Nottingham NG7 2RD, UK; joanne.ablewhite@nottingham.ac.uk (J.A.); m.orrell@nottingham.ac.uk (M.O.); 2Cambridge Institute for Music Therapy Research, Anglia Ruskin University, Cambridge CB1 1PT, UK; jodie.bloska@aru.ac.uk (J.B.); helen.odell-miller@aru.ac.uk (H.O.-M.); 3Health & Social Sciences, NORCE Norwegian Research Centre, 5008 Bergen, Norway; jorg.assmus@uni.no (J.A.); chgo@norceresearch.no (C.G.); visv@norceresearch.no (V.S.)

**Keywords:** depression, dementia, music, care home, RCT, music therapy, choir singing

## Abstract

**Background:** We report UK findings from Music Interventions for Depression and Dementia in Elderly care (MIDDEL), a cross-national, clustered, randomised trial undertaken in 2018–2023 to evaluate the effectiveness of music interventions for depression symptoms in care home residents living with dementia (NCT03496675, clinicaltrials.gov (accessed on 1 December 2024)). The trial compared the effects of Group Music Therapy (GMT) with Recreational Choir Singing (RCS); GMT and RCS combined; and treatment as usual (TAU). **Methods:** In the intervention arms, the protocolized music interventions were delivered in care home units twice per week for three months, then once per week for three months. The primary outcome was depressive symptoms after six months, measured by MADRS. Secondary outcomes included well-being—EQ-5D-5L, Visual Analogue Scale (VAS); quality of life—QOL-AD; symptoms of dementia—SIB-8, NPI-Q; and caregiver distress—NPI-Q. The change in MADRS score from baseline to 6 months was assessed using a linear mixed-effects model. We report the multivariate model having both treatments as predictors, both unadjusted and adjusted, for the interaction between the treatments. **Results:** The UK trial started in 2022 after the pandemic lockdown, when 16 care home units were recruited and randomised, four per arm; 192 residents aged over 65 with depression and dementia participated. An ITT analysis of 146 participants retained at 6 months found neither intervention had a significant positive effect on any outcome. Significant unfavourable effects were found for RCS participants on MADRS, NPI symptom severity, and EQ-VAS. The combination of RCS + GMT had a detrimental effect on caregiver distress. **Conclusions:** MIDDEL UK findings do not support the use of GMT or RCS to alleviate depression in care home residents with dementia.

## 1. Introduction

This paper reports the UK results from Music Interventions for Depression and Dementia in Elderly care (MIDDEL), a large international research project that investigated the effect of music interventions in reducing depressive symptoms among elderly people with dementia and depressive symptoms living in care homes. By focussing on one country, we hope to present contextualised findings that can be applied judiciously by those familiar with UK care homes.

Dementia and depression are relatively common conditions among older adults, entailing great challenges for those affected, including relatives and care personnel, and significant costs for society. However, music-based interventions represent a very promising form of intervention, both for dementia and for depression among older people. The use of such interventions is widespread but has not yet been thoroughly investigated in large studies. Music has the advantages of being easily available, low-cost, and popular in most societies. As to the evidence base, undifferentiated music-based interventions, including singing, are consistently judged favourably in relation to dementia-related symptoms of depression [1,2], agitation [3], anxiety [4], and cognitive function [5]. However, the evidence is piecemeal, and most reviews conclude with a call for more research.

The MIDDEL cross-national study was implemented in Australia, the Netherlands, Norway, Turkey, Germany, and the UK from 18 July 2018 to 31 December 2023, with over 1000 participating residents distributed across 86 care home units [6,7]. The UK part of the study involved 192 residents in 16 care home units, with data collected in 2022–2023. MIDDEL constitutes the largest study to date of music interventions in residents with depression and dementia residing in care homes and the first to compare the effects of two different music interventions—Group Music Therapy (GMT) and Recreational Choir Singing (RCS).

The main aim of MIDDEL was to provide high-quality evidence on the comparative effectiveness of GMT, RCS, GMT in combination with RCS, or usual care in reducing depressive symptoms among care home residents with dementia. Secondary objectives were to examine the effects of GMT and RCS on cognitive function, neuropsychiatric symptoms, quality of life, dementia severity, functional impairment, and mortality in the residents, in addition to workload factors affecting staff at the care home units. The protocol for the trial (NCT03496675 at https://clinicaltrials.gov/ (accessed on 20 May 2025)) details the design and methods [8]. The following amendments were made after the publication of the protocol: inclusion criteria were expanded to a Clinical Dementia Rating (CDR) score of 0.5–3 (very mild to severe) to include a broader spectrum of people with dementia; CDR was excluded as a secondary outcome at follow-up due to the burden on participants and limited expectations for change; and the Severe Impairment Battery (SIB-8) was added as a secondary outcome measure of cognitive impairment in Europe, as it was considered more sensitive to changes in cognition over time.

## 2. Materials and Methods

MIDDEL was a pragmatic, four-arm, cluster-randomised controlled trial that used a 2 × 2 factorial design to compare the effects of GMT, RCS, GMT and RCS combined, or usual care for care home residents with both dementia and depressive symptoms. A factorial design was used to investigate the effectiveness of two group music interventions, as well as the combined and interaction effects.

### 2.1. Participants

Eligibility for the study was defined for both care home units and individual participants. Care home units were invited to participate if they had at least 10 eligible and consenting residents. Care homes were excluded if they were currently providing music-based interventions as part of their usual care program. Eligibility criteria for resident participants included the following: aged 65 years or older; resident (full-time, 24 h/day) at a participating care home; dementia as indicated by a Clinical Dementia Rating score of 0.5 to 3 [9] and a Mini-Mental State Examination (MMSE) score of 26 or less [10]; at least mild depressive symptoms, as indicated by a Montgomery-Åsberg Depression Rating Scale (MADRS) score of at least 8 [11]; a clinical diagnosis of dementia according to ICD-10 research criteria; and to have given written informed consent or assent by proxy for those unable to provide consent themselves. Because most (but not all) residents lacked the ‘mental capacity’ for responding, care staff who knew them well acted as informants.

### 2.2. Experimental Design and Procedures

The care home units were recruited and randomised into four groups: care home units receiving GMT; RCS; both; or a control group that received usual care without music intervention. A core principle of Group Music Therapy (GMT) is to regulate affect through active, reciprocal music making with the use of singing and musical instruments. This facilitates the relationship between the music therapist and the person living with dementia, and between participants in the group. Another core principle of GMT is to meet the psychosocial needs of each individual resident, which, in turn, is thought to reduce depressive symptoms and anxiety and to stimulate overall social and emotional well-being. GMT is provided by a trained music therapist who is registered with the appropriate professional association or registration body in their country and is a skilled musician and therapist. The group size is often restricted to no more than 8 [12].

By comparison, the approach of Recreational Choir Singing (RCS), facilitated by choir leaders for whom formal training is not required, is to sing familiar songs and to provide a lively musical environment for participants. RCS involves a combination of cognitive, physical, and psychosocial engagement components. Biographically and culturally grounded song materials are used with the central goal of stimulating positive experiences shared by groups of individuals. As applied in this trial, RCS aims to foster connectedness, emotional well-being, and enjoyment of music-making in a group where larger numbers can be included than in GMT.

UK COVID-19 lockdown restrictions meant that the study could not start delivering interventions until early in 2022, and data collection continued until the end of 2023. Each intervention session was 45 min long. The residents in the three experimental arms received interventions over 6 months, of which the first 3 months consisted of two sessions per week per form of intervention, and the following 3 months consisted of weekly sessions per intervention. Thus, four homes received two sessions of GMT plus two sessions of RCS: three hours of intervention time per week for the first three months. Four homes received GMT, four received RCS, and four received no intervention. Music therapists and choir leaders were recruited and trained according to manuals for implementation that were developed as part of the cross-national study and adapted for the UK arm. Interventionists kept registers and completed fidelity checklists at the end of each session, reviewing sessions for fidelity in regular supervision sessions. Outcomes were measured at months 0 (baseline), 3, and 6 (primary time point). See the RCS and GMT Handbooks in Appendix A; these contain the fidelity checklists.

### 2.3. Randomisation and Blinding 

Block randomisation (block size of four) was used to ensure that each site would have a balanced distribution between the interventions. The randomisation of CHUs was performed by CG using a computer-generated randomisation list after baseline assessments had been completed and entered into the electronic database. Where possible, four units were randomised at a time to ensure allocation concealment.

Due to the nature of the interventions, care staff, intervention providers, and residents were not blinded to the intervention provided. However, all assessments were performed by trained clinical researchers who were kept in ignorance of the allocation status of participants. The success of blinding was determined at each follow-up assessment by asking assessors if they had inadvertently become aware of the allocation of the participant. One case of unblinding was reported at 3 months, and the assessor was replaced to ensure masking for the remaining assessments. There were no cases of unblinding reported at 6 months.

### 2.4. Measurement Tools

The primary outcome was the MADRS measured after 6 months of the music interventions. The MADRS measures depressive symptoms using a 10-item scale, where each item is rated from 0 (not present) to 6 (severe). The total score ranges from 0–60, with higher scores indicating higher severity of depressive symptoms. The MADRS was completed by self-report or by a proxy staff member involved in the residents’ care, which is recommended where participants are unable to provide definitive responses themselves [13].

As for secondary outcomes, the Severe Impairment Battery-8 (SIB-8) [14] was chosen to measure cognitive impairment because it is sensitive to changes in impairment over time. Its score is derived from a semi-structured interview with the person living with dementia and an appropriate caregiver/relative. It rates impairment in each of 6 cognitive categories (memory, orientation, judgment and problem solving, community affairs, home and hobbies, and personal care). The Neuropsychiatric Inventory Questionnaire (NPI-Q) was used to measure the severity of neuropsychiatric symptoms and has substantial evidence of validity and reliability [15,16]. The NPI-Q measures neuropsychiatric symptoms across 12 symptom domains, where if a symptom is present, it is rated by proxy care staff on two subscales: the symptom’s severity (from 1 [mild] to 3 [severe]) and the associated distress for caregivers (from 0 [not distressing at all] to 5 [extremely distressing]).

Quality of life was measured using EuroQol (EQ-5D-5L), a generic health utility measure suitable for people with dementia [17] that is used to derive quality-adjusted life-years (QALYs). It is based on a descriptive system that defines health in the five dimensions: mobility, self-care, usual activities, pain/discomfort, and anxiety/depression. Each dimension has five response categories, from “no problems” to “extreme problems”, which are combined using preference weights to form an overall quality of life score ranging from lower than 0 (worse than death) to 1 (best possible). An additional Visual Analogue Scale (VAS) indicates self- or proxy-reported health on a scale from 0 (“The worst health you can imagine”) to 100 (“The best health you can imagine”).

Disease-specific quality of life was measured using the Quality of Life in Alzheimer’s Dementia (QOL-AD) measure [18]. The QOL-AD is a 13-item scale with a self-rating and proxy version that can be used by people with very low MMSE scores. Items such as “Physical health”, “Memory”, or “Ability to do things for fun” are scored on a scale from 1 (poor) to 4 (excellent), resulting in a total score ranging from 13 (worst) to 52 (best). As most residents were unable to self-rate the EQ-5D-5L and the QOL-AD, the proxy ratings are deemed most reliable in this study.

A data and study monitoring committee (DSMC) with unblinded access to the study data reviewed biannual updates on adverse events (hospitalisations and all-cause mortality) during the project period. No related serious adverse events were reported, and hospitalisation rates were similar across groups.

### 2.5. Analysis

The analysis was performed using the intention-to-treat (ITT) approach, to use all available data from all participants as randomised, regardless of the intervention received. Additional per-protocol analysis (PPA) was performed on those participants who received at least 50% of their intended music interventions. The general significance level was set to 0.05, but since there are two comparisons in the primary analysis (GMT vs. no GMT, RCS vs. no RCS) a marginal Bonferroni level of 0.025 was used to indicate a non-random result on the primary outcome, which is a change in MADRS score from baseline to 6 months. All computations were performed using R (R Core Team, v 4.1.2, 2021). Sociodemographic and clinical baseline properties for the groups were characterised by descriptive methods (mean [SD], median [range] or *n* [%]). Those who dropped out versus those who completed the intervention were compared. The primary outcome was assessed by a linear mixed-effects model (LME). We fitted the unadjusted model for each treatment (RCS vs. no RCS), as well as the multivariate model having both treatments, as predictors—both unadjusted and adjusted—for the interaction between the treatments.

## 3. Results

### 3.1. Baseline Measures

Table 1 shows the characteristics of the participants at baseline, by study arm: Study Control (SC) GMT, RCS, and GMT + RCS. Due to the memory impairment of the participating residents, information about education was not gathered. All had a diagnosis of dementia confirmed by their MMSE and CDR scores, but specific details of the diagnosis were not known. The mean age was 86, and 75% of participants were female. The participant profile is a moderately to severely disabled population of people with dementia. This is fairly typical of UK care home residents, where the policy is to delay admission to long-term care until there is no alternative. There was no significant variation between the four arms of the trial, but there appears to be a trend towards the RCS participants being more cognitively impaired.

The CONSORT diagram (Appendix A) shows that out of 192 participants, 161 residents were available for interview after three months (84%), and 146 residents were followed up at six months (76%), the study end point. Between the baseline and the 6-month assessment, there were 22 deaths (11.5%). Seventeen people left the care home where they had been recruited (9%), four participants withdrew from the study, and two were too unwell to carry on. Data were collected from care personnel on certain measures, so the number of participants on these variables exceeds 146.

### 3.2. Outcomes

The primary outcome measure was depression in dementia at 6 months post-baseline, as measured by the MADRS. Figure 1 summarises the results of ITT analyses for this and the secondary variables graphically, and Table 2 reports the results of the linear mixed-effects modelling on all six outcomes.

Focussing on the ANCOVA results for the full model and taking Bonferroni-adjusted *p* < 0.025 as the threshold of probability, statistically significant unfavourable effects were found for the participants in the RCS group on the MADRS; RCS was associated with significantly higher ratings for depression symptoms (*p* = 0.003). This was not the case for GMT nor for the combination of RCS + GMT (Table 2a).

No significant differences were found between intervention and control groups in relation to cognitive impairment (SIB-8, Table 2b). RCS was associated with higher NPI symptom severity (*p* = 0.008, Table 2c). The interaction of GMTxRCS was associated with higher caregiver distress (*p* = 0.002, Table 2d). Participants in RCS had lower self-rated quality of life on the EQ-VAS (*p* = 0.009, Table 2e). However, the interaction GMTxRCS was associated with a higher EQ-VAS, although only marginally significant (*p* = 0.024, Table 2e).

The ANCOVA full model results for QOL-AD (Table 2f) are notable because, as independent predictors, GMT (*p* = 0.032) and RCS *p* = 0.026) were each associated with improvements in quality of life, even though these associations are only near-significant given the Bonferroni-adjusted threshold of *p* < 0.025. It is striking, therefore, that the interaction of GMTxRCS was significantly associated with a lower quality of life (*p* = 0.016, Table 2f).

### 3.3. Per-Protocol Analysis

The per-protocol analysis of data from 84 participants who completed at least 50% of their allocated interventions presents a very similar picture with regard to ANCOVA: the RCS group had a significantly higher MADRS (depression) score, higher (non-significant) NPI-severity score, and worse (non-significant) cognitive impairment as measured by the SIB-8 score, while the GMT group showed a similar but non-significant trend on all three outcomes. The interaction RCSxGMT was associated with a worse NPI-caregiver distress score (Appendix A).

## 4. Discussion

The results do not align with previous indications that music-based interventions can reduce symptoms of depression in dementia [2,19,20]. Moreover, RCS seemed to be associated with worsening outcome measures. By comparison with RCS, GMT was not significantly associated with poorer outcomes. In relation to caregiver distress, the combination of RCS + GMT had a negative effect, although each intervention alone had near-significant positive effects. In relation to quality of life (QOL-AD), a similar pattern appears: the interaction of RCSxGMT had a negative effect (*p* = 0.016), although the individual interventions point to a positive effect.

These surprising findings need to be interpreted to be understood. The impact of the COVID-19 pandemic on the study is explored elsewhere [21]. Briefly, this context generated impediments to staff engagement with the study and logistical issues due to virus control measures as care homes emerged from lockdown. One account of our results is simply that RCS (singing in groups for 45 min once or twice a week) was detrimental to this frail, elderly population who had recently experienced lockdown in residential care. That could be due to the perceived upheavals and disruption associated with the intervention. By comparison, GMT, with its music therapy-informed, individualised approach, may have mitigated the negative impact so that the results were not statistically significant.

Negative results were indicated for people receiving the combination of both types of intervention, which amounted to 4 × 45 min sessions per week in the first three months. One explanation may be that residents disliked being required to attend groups in their own homes, and that very frequent sessions were unwelcome because of their impact on routines, resulting in negative perceptions. Alternatively, it has been found that sad music increases symptoms in young people who have tendencies to depression [22]—perhaps there is a similar response in some of the study participants, which could offset benefits to others.

Another possible explanation is bias from proxy informants. Care home personnel responded to the MADRS on behalf of all but two residents (99%), 82% of the participants for the EQ-5D, and 64% for the QOL-AD. Given staffing rotas, sickness absences, and the fact that staff move on from care homes at a steady rate, it is likely that different people rated any given participant at baseline and follow-up, using varying perceptions of that resident; still, the bias in this case would be random.

In some instances, it may be argued that staff were biased in their scoring because they were aware of the intervention status of the participants (i.e., control or experimental arm). Care home personnel could not fail to be aware that a music intervention was or was not being delivered frequently in their workplace. Again, bias could go in either direction: a member of staff might want to show their care home in a positive light, perhaps particularly in the control arm, because UK care homes operate in a competitive marketplace. Alternatively, given the stressful context of their job, personnel might resent the demands of the research, which sometimes increased the carer burden, making their responses careless or unreliable.

A further explanation is that the treatment as usual in the ‘control’ homes was in some way more effective than the intervention. Whilst homes delivering regular music interventions were ineligible, treatment as usual in control homes may have incorporated ad hoc access to music, live and recorded. All homes were provided with information prior to entering the study and, therefore, would know that residents living in those care homes in the study that were randomised to receive the intervention would be receiving regular music sessions. It is possible that the control homes introduced music or alternative active interventions once the study was underway, which yielded exceptional benefits for their residents in the TAU arm, reducing the differences found at six months.

As evidence of engagement with the research in certain participating homes, at least three of them invited the interventionists back to deliver more music after the study had ended. By contrast, it was clear to the researchers that certain homes did not embrace the study—no doubt due to other urgent priorities. In an exploratory post hoc analysis, we compared outcomes for homes grouped according to high (4), medium (9), or low (3) engagement with the study. No clear differences in outcomes emerged; however, the analysis may be underpowered.

### 4.1. Limitations

This was a pragmatic trial undertaken immediately after a pandemic disrupted UK residential care facilities even more than it affected the general population [21]. The music-based interventions offered under the constraints of an RCT may have increased the burden on care personnel and residents. There was considerable dropout from the allocated interventions. The turnover of personnel may have affected the reliability and validity of their proxy responses. Virus control restrictions still in operation meant that training was completed online rather than face-to-face, which may have impaired the fidelity of the research process.

### 4.2. Future Research

Much of the evidence for singing in dementia derives from non-randomised studies, where participants effectively self-selected. Few RCTs of music-based interventions have been completed in care home settings [23], but one systematic review of music-based RCTs points to the importance of active engagement with music-based interventions, including singing [20]. The pursuit of engagement is a feature of music therapy-informed practice. It will be manifested in different ways by people with different levels of dementia. Therefore, we suggest that residents should be screened for their affinity with group singing (or other music-based interventions), including their capability to participate actively in the intervention on offer. In addition, measurement of active engagement implies a need for observational ‘in the moment’ evaluation approaches, including observational methods and biometrics such as salivary cortisol, heart rate, or galvanic skin response.

Similarly, music interventionists need to be skilled and fully supported to work with older people who are disabled by dementia and depression in care settings. This implies knowledge and understanding of the participants and the disorders, training in using music-based activities with this population, confidence gained through experience, and support through supervision. Advice from staff who know the residents is indispensable, and the environment where the intervention is delivered should be suitable and accommodate it adequately. Complex interventions such as music call for the thorough preparation of all staff, participants, and visitors in care settings. In responding to calls for more randomised trials of music interventions [24], researchers and, particularly, methodologists and reviewers of research protocols, need to understand the obstacles to research that can affect residential settings.

## 5. Conclusions

The findings of the MIDDEL study in the UK do not support either RCS or GMT being labelled as a therapeutic intervention to decrease depressive symptoms for care home residents with dementia. These results challenge prevailing beliefs on the general efficacy of music-based interventions in dementia and depression by pointing to the importance of tailoring interventions to the capabilities and needs of individuals. From a methodological perspective, the accomplishment of a large RCT of music-based interventions in UK care homes has highlighted many aspects of the research process that require forethought and planning if future research of this kind is to succeed.

## Figures and Tables

**Figure 1 geriatrics-10-00166-f001:**
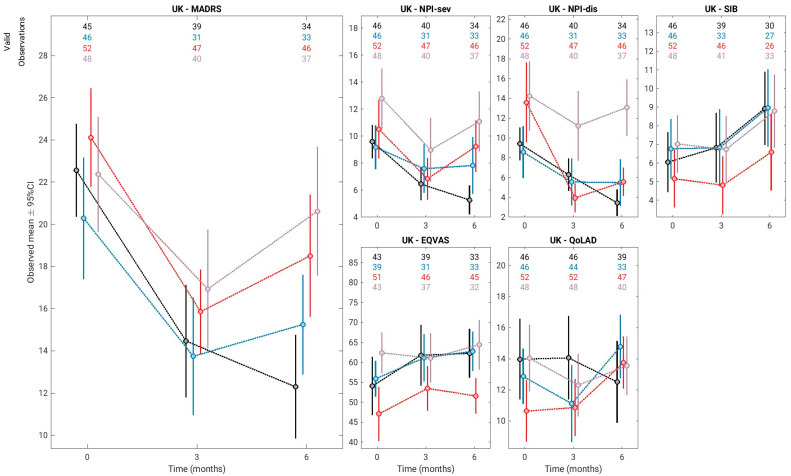
ITT analysis of all outcomes (black for TAU, blue for GMT, red for RCS and grey for RCS + GMT).

**Table 1 geriatrics-10-00166-t001:** Baseline Characteristics of MIDDEL-UK Participants with Depression and Dementia.

	Total Sample		Study Control/TAU		GMT		RCS		GMT + RCS	
	*n* = 192		*n* = 46		*n* = 46		*n* = 52		*n* = 48	
	Valid	Value	Valid	Value	Valid	Value	Valid	Value	Valid	Value
**Age (years) ^1^**	192	85.8 (7.6)	46	86.2 (6.9)	46	86.9 (7.6)	52	85.5 (8.1)	48	84.7 (7.8)
**Sex (Female)**	192	144 75%	46	29 63%	46	38 82.6%	52	41 78.8%	48	36 75%
**MADRS** **(0–60) ^1^**	191	22.4 (9)	45	22.6 (7.6)	46	20.3 (10)	52	24.1 (8.6)	48	22.4 (9.6)
**MMSE** **(0–30) ^1^**	191	7.8 (7.6)	46	7.6 (8)	46	8.8 (7.4)	51	6.1 (7.1)	48	8.8 (7.9)
**CDR** **(0.5–3) ^1^**	192	2.3 (0.7)	46	2.2 (0.8)	46	2.1 (0.8)	52	2.7 (0.6)	48	2.2 (0.6)
**SIB-8 ^1^**	192	6.2 (5.6)	46	-	46	6.8 (5.5)	52	5.2 (5.7)	48	7 (5.5)
**NPI Severity** **(0–36) ^1^**	192	10.5 (6.8)	46	9.6 (4.3)	46	9.2 (5.6)	52	10.5 (8)	48	12.8 (7.9)
**NPI Dis** **(0–60) ^1^**	192	11.6 (11.4)	46	9.4 (5.8)	46	8.6 (9.1)	52	13.6 (14.8)	48	14.2 (12.4)
**EQ-VAS** **(0–100) ^1^**	176	54.5 (21.6)	43	54.1 (24.5)	39	55.9 (14.3)	51	47.1 (24.7)	43	62.4 (17.1)

^1^ Mean (SD).

**Table 2 geriatrics-10-00166-t002:** ITT analysis of primary and secondary outcomes *.

**(a) MADRS (Primary outcome).**					
		ANCOVA		ANOVA	
	N	B (95% CI)	*p*-value	*p*-value	*p*-value
RCS only	149	5.5 (2.7, 8.4)		<0.001	
RCS			<0.001		
GMT only	149				0.016
GMT		2.7 (0.3, 5.6)	0.077		
RCS and GMT	149			0.01	<0.001
RCS		5.6 (2.8, 8.5)	<0.001		
GMT		2.9 (0.1, 5.7)	0.043		
Full model	149			0.045	0.045
RCS		6.0 (2.1, 9.9)	0.003		
GMT		3.3 (−0.9, 7.5)	0.118		
RCS × GMT		−0.8 (−6.4, 4.8)	0.781		
**(b) Severe Impairment Battery, SIB-8.**					
		ANCOVA		ANOVA	
	N	B (95% CI)	*p*-value	*p*-value	*p*-value
RCS only	116			0.082	
RCS		−1.0 (−2.6, 0.6)	0.208		
GMT only	116				0.08
GMT		1.0 (−0.6, 2.6)	0.202		
RCS and GMT	116			0.066	0.068
RCS		−1.1 (−2.7, 0.5)	0.168		
GMT		1.1 (−0.5, 2.7)	0.164		
Full model	116			0.049	0.049
RCS		−2.0 (−4.3, 0.3)	0.09		
GMT		0.3 (−2, 2.6)	0.804		
RCS × GMT		1.7 (−1.5, 4.9)	0.303		
**(c) Neuropsychiatric Inventory (NPI) Symptom severity.**					
		ANCOVA		ANOVA	
	N	B (95% CI)	*p*-value	*p*-value	*p*-value
RCS only	150			0.001	
RCS		3.2 (1.3, 5.2)	0.001		
GMT only	150				0.024
GMT		1.8 (−0.2, 3.8)	0.071		
RCS and GMT	150			0.014	<0.001
RCS		3.3 (1.4, 5.3)	0.001		
GMT		2.0 (0.1, 3.9)	0.04		
Full model	150			0.071	0.071
RCS		3.6 (1, 6.3)	0.008		
GMT		2.3 (−0.5, 5.2)	0.107		
RCS × GMT		−0.6 (−4.5, 3.2)	0.749		
**(d) Neuropsychiatric Inventory (NPI) Caregiver distress.**					
		ANCOVA		ANOVA	
	N	B (95% CI)	*p*-value	*p*-value	*p*-value
RCS only	150			<0.001	
RCS		3.8 (1.6, 6)	0.001		
GMT only	150				<0.001
GMT		5.0 (2.9, 7.1)	<0.001		
RCS and GMT	150			<0.001	<0.001
RCS		4.0 (2, 6.1)	<0.001		
GMT		5.2 (3.2, 7.2)	<0.001		
Full model	150			<0.001	<0.001
RCS		1.0 (−1.8, 3.7)	0.487		
GMT		1.7 (−1.2, 4.6)	0.251		
RCS × GMT		6.4 (2.4, 10.3)	0.002		
**(e) Quality of Life: EQ-VAS.**					
		ANCOVA		ANOVA	
	N	B (95% CI)	*p*-value	*p*-value	*p*-value
RCS only	134			0.011	
RCS		−4.6 (−10.4, 1.1)	0.114		
GMT only	134				0.003
GMT		6.7 (0.8, 12.6)	0.027		
RCS and GMT	134			0.003	0.014
RCS		−4.2 (−9.9, 1.5)	0.146		
GMT		6.4 (0.5, 12.3)	0.034		
Full model	134			0.001	0.001
RCS		−10.2 (−17.8, −2.5)	0.009		
GMT		−0.4 (−8.7, 7.9)	0.922		
RCS × GMT		13.4 (1.8, 25)	0.024		
**(f) Quality of Life: QOL-AD.**					
		ANCOVA		ANOVA	
	N	B (95% CI)	*p*-value	*p*-value	*p*-value
RCS only	159			0.125	
RCS		0.7 (−1.1, 2.6)	0.436		
GMT only	159				0.154
GMT		0.5 (−1.4, 2.4)	0.592		
RCS and GMT	159			0.155	0.125
RCS		0.7 (−1.2, 2.6)	0.44		
GMT		0.5 (−1.4, 2.4)	0.598		
Full model	159			0.003	0.003
RCS		2.9 (0.4, 5.5)	0.026		
GMT		3.0 (0.3, 5.8)	0.032		
RCS × GMT		−4.7 (−8.4, −0.9)	0.016		

* Shaded rows highlight the full models for ease of comparison.

## Data Availability

De-identified datasets (participant codes and outcome scores) generated during and/or analysed during the MIDDEL trial are stored in the publicly available repository OSF and can be accessed at https://doi.org/10.17605/OSF.IO/TPDHU (accessed on 1 December 2024).

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
