# Peer review of "Impact of Music Interventions on Depression in Care Home Residents with Dementia: UK Results from Music Interventions for Depression and Dementia in Elderly Care RCT"

_geriatrics, 2025, doi:10.3390/geriatrics10060166_

Round 1

Reviewer 1 Report

Comments and Suggestions for Authors

The paper deals with the effects of music interventions on symptoms of depression in care home residents with dementia. First, I would like to point out that it is very important that study results that are insignificant or that contradict the formulated hypotheses are also published. As this happens so rarely, many research areas are subject to the well-known phenomenon of publication bias. This paper is therefore an interesting contribution to the research field of music interventions in dementia, especially because the results are so surprising. However, I have a few comments that should be clarified before the paper is published.

Introduction

The introduction could stand to benefit from a brief section on the evidence in favor of music therapy and group singing in care homes, particularly in people with dementia (and depression). The most important studies on which the hypotheses of the present study are based should be cited.

Methods

Did the care staff receive any training in the use of the scales and questionnaires? The use of the MADRS in particular requires sound practical expertise with the instrument due to the differentiated gradations in the assessment of symptoms. The same applies to the use of the NPI. If no appropriate training and supervision for quality assurance for proxy-assessments has taken place here, this would be a clear limitation of the study and should be discussed.

Discussion

Might pre-post designs be less suitable or not sensitive enough to assess the effects of music therapy in people with advanced dementia and should situational or process-based evaluation methods be used in addition or instead (e.g. observational and rating scales)? Please discuss these aspects with regard to future research (e.g. the development of more appropriate instruments and study designs).

Furthermore, studies have shown that family or professional caregivers often rate the quality of life in people with dementia lower than individuals with dementia themselves. Could this have had an influence on the results? And what was the general level of agreement between self-reporting and proxy-assessment of quality of life?

In view of the results of the study, the fundamental question arises of whether group music interventions are still suitable at all for people with severe dementia. Studies rather show that an individualized music therapy approach is more effective for advanced dementia. This should also be discussed based on the relevant evidence.

Author Response

Thank you for the careful and thorough review.

Comment 1: The introduction could stand to benefit from a brief section on the evidence in favor of music therapy and group singing in care homes, particularly in people with dementia (and depression). The most important studies on which the hypotheses of the present study are based should be cited.

Response 1: We have added the following, together with five references to recent reviews at line 46. Music has the advantages of being easily available, low cost and popular in most societies.  As to the evidence base, undifferentiated music-based interventions including singing are consistently judged favorably in relation to dementia-related symptoms of depression [1,2], agitation [3], anxiety [4] and cognitive function [5]. However, most reviews conclude with a call for more research. 

Comment 2: Did the care staff receive any training in the use of the scales and questionnaires? The use of the MADRS in particular requires sound practical expertise with the instrument due to the differentiated gradations in the assessment of symptoms. The same applies to the use of the NPI. If no appropriate training and supervision for quality assurance for proxy-assessments has taken place here, this would be a clear limitation of the study and should be discussed.

Response 2: The data collection was done not by care staff but by trained clinical researchers who were kept blind to participants' intervention status.  This is not clear, so we have added the following at line 113:

All data was collected by trained clinical researchers who were kept in ignorance of the allocation status of participants. Because most (but not all) residents lacked ‘mental capacity’ for responding, care staff who knew them well acted as informants. 

Comment 3: Might pre-post designs be less suitable or not sensitive enough to assess the effects of music therapy in people with advanced dementia and should situational or process-based evaluation methods be used in addition or instead (e.g. observational and rating scales)? Please discuss these aspects with regard to future research (e.g. the development of more appropriate instruments and study designs).

Response 3:  We are conscious of the length of this paper and the additions already made so, for the sake of brevity, at line 288 we have added the following:  The measurement of active engagement implies a need for observational, ‘in the moment’ evaluation approaches including biometrics such as salivary cortisol or galvanic skin response." rather than embarking on a full discussion here. 

Comment 4: Furthermore, studies have shown that family or professional caregivers often rate the quality of life in people with dementia lower than individuals with dementia themselves. Could this have had an influence on the results? And what was the general level of agreement between self-reporting and proxy-assessment of quality of life?

Response 4: It is true that QOL assessment show this divergence. For that reason we used the MADRS as our primary outcome. Unfortunately, there were only a handful of participants who could rate their own wellbeing.  Therefore this consideration, while important in general, would not affect our results. 

Comment 5: In view of the results of the study, the fundamental question arises of whether group music interventions are still suitable at all for people with severe dementia. Studies rather show that an individualized music therapy approach is more effective for advanced dementia. This should also be discussed based on the relevant evidence.

Response 5:  The question raised is interesting, however we consider that the issue of group versus individual interventions lies beyond the scope of this paper. The MIDDEL study did not generate results about the latter and we are not aware of conclusive studies which we could bring to this discussion showing that individualised music therapy is more effective than group music therapy (which was one of our interventions) or more effective than another music-based intervention for advanced dementia. Again, the need to restrict additions to the paper's length inclines us to suggest that this is a discussion for another place. 

Reviewer 2 Report

Comments and Suggestions for Authors

This study examined the effects of group music therapy (GMT), recreational choir singing (RCS), and the combined application of the two interventions on depression in dementia patients residing in care homes. However, I think that there are many things that need to be improved in order for the submitted manuscript to be published in journal. You need to supplement the following.

1. When an abbreviation appears first in the abstract, the full name should be written.

 2. If you want to verify the effectiveness, you must present a rationale for why GMT or RCS may be effective in the introduction and provide logical validity for the research design..

 3. In the Materials and Methods section, please provide specific information, organized into subheadings, about participants, measurement tools, experimental design and procedures, and analysis methods. 

4. When you describe the procedure, you should be specific about participant allocation (whether the RCT was successful), ethical considerations, etc.

5. And most importantly, you should describe in detail the components of the group program you applied, and also explain specifically how RCS was standardized and applied.

6. Please edit Table 2 to make it easier to read.

7. Please discuss the results specifically based on the comments I made above.

Author Response

Thanks for this very helpful review. 

Comment 1. When an abbreviation appears first in the abstract, the full name should be written.

Response 1: Thanks for pointing this out.  We have added the study title in full to the abstract at line 12:

Music Interventions for Depression and Dementia in Elderly care (MIDDEL), 

 2. If you want to verify the effectiveness, you must present a rationale for why GMT or RCS may be effective in the introduction and provide logical validity for the research design..

Response 2:  The rationale for why GMT, RCS or any music intervention may be effective has now been specified more fully, in relation to research evidence, with the addition of the following justification including 5 references at line 46ff.

As to the evidence base, undifferentiated music-based interventions including singing are consistently judged favorably in relation to dementia-related symptoms of depression [1,2], agitation [3], anxiety [4] and cognitive function [5]. However, most reviews conclude with a call for more research.  

We hope that this summary of the evidence also provides justification for the research design, which is described in the following paragraph, including "MIDDEL constitutes the largest study to date of music interventions in residents with depression and dementia residing in care homes, and the first to compare the effects of two different music interventions – group music therapy and recreational choir singing."

 3. In the Materials and Methods section, please provide specific information, organized into subheadings, about participants, measurement tools, experimental design and procedures, and analysis methods. 

Response 3: These subheadings have been added.

4. When you describe the procedure, you should be specific about participant allocation (whether the RCT was successful), ethical considerations, etc.

Response 4:  Not entirely sure what is meant here, but we have added the following at line 114:

All data was collected by trained clinical researchers who were kept in ignorance of the allocation status of participants. Because most (but not all) residents lacked ‘mental capacity’ for responding, care staff who knew them well acted as informants. 

5. And most importantly, you should describe in detail the components of the group program you applied, and also explain specifically how RCS was standardized and applied.

Please see lines 87 to 103.  Line 110 continues "Music therapists and choir leaders were recruited and trained according to manuals for implementation that were developed as part of the cross-national study and adapted for the UK arm." We go on to state that the intervention manuals are provided as supplementary material. 

6. Please edit Table 2 to make it easier to read. 

Response 6:  Done, hope this is better. 

7. Please discuss the results specifically based on the comments I made above.

We'd welcome further clarification of this comment because if we have understood the comments they apply to the formal parts of the paper rather than to the interpretation of the findings. 

Round 2

Reviewer 2 Report

Comments and Suggestions for Authors

Thank you for improving your manuscript based on my comments in the first review. I think it would be good if you put a little more effort into writing the manuscript.

1. First, please proofread the original text a little more carefully and improve readability.

2. You need to be a little more careful when you make the table. For example, in a paper, you need to be a little more restrained when writing the line. Also, in the case of the table 2, the tables are divided, but they are presented under one table title.

3. Please describe the limitations of this study in detail. 

4. The conclusion needs to be written in a little more detail

Author Response

  1. First, please proofread the original text a little more carefully and improve readability. 

Done, thank you.

2. You need to be a little more careful when you make the table. For example, in a paper, you need to be a little more restrained when writing the line. Also, in the case of the table 2, the tables are divided, but they are presented under one table title.

I have removed the horizontal lines.  I am reluctant to alter the numbering of Table 2 because this corresponds to the similar table in the main paper. 

3. Please describe the limitations of this study in detail. 

Done - see paragraph that starts at line 306:

Limitations

This was a pragmatic trial undertaken immediately after a pandemic disrupted UK residential care facilities even more than it affected the general population [21]. The music-based interventions offered under the constraints of an RCT may have increased the burden on care personnel and residents. There was considerable drop out from the allocated interventions.  The turnover of personnel may have affected the reliability and validity of their proxy responses. Virus control restrictions still in operation meant that training was completed online rather than face to face which may have impaired the fidelity of the research process.

4. The conclusion needs to be written in a little more detail

I have elaborated a little, mindful of length. as follows: 

Conclusions

The findings of the MIDDEL study in the UK do not support either RCS or GMT being labelled as a therapeutic intervention to decrease depressive symptoms for care home residents with dementia. These results challenge prevailing beliefs in the general efficacy of music-based interventions in dementia and depression, by pointing to the importance of tailoring interventions to the capability and needs of individuals. From a methodological perspective, the accomplishment of a large RCT of music-based interventions in UK care homes has highlighted many aspects of the research process that require forethought and planning if future research of this kind is to succeed.